# SCRMshaw: Supervised *cis*-regulatory module prediction for insect genomes

Hasiba Asma[1], Luna Liu[2], Marc S. Halfon[1,2,3]*

1 Departments of Biochemistry, University at Buffalo-State University of New York, Buffalo, NY, United States of America, 2 Biomedical Informatics, University at Buffalo-State University of New York, Buffalo, NY, United States of America, 3 Biological Sciences, University at Buffalo-State University of New York, Buffalo, NY, United States of America

* mshalfon@buffalo.edu

## Abstract

As the number of sequenced insect genomes continues to grow, there is a pressing need for rapid and accurate annotation of their regulatory component. SCRMshaw is a computational tool designed to predict *cis*-regulatory modules ("enhancers") in the genomes of various insect species. A key advantage of SCRMshaw is its accessibility. It requires minimal resources—just a genome sequence and training data from known *Drosophila* regulatory sequences, which are readily available for download. Even users with modest computational skills can run SCRMshaw on a desktop computer for basic applications, although a high-performance computing cluster is recommended for optimal results. SCRMshaw can be tailored to specific needs: users can employ a single set of training data to predict enhancers associated with a particular gene expression pattern, or utilize multiple sets to provide a first-pass regulatory annotation for a newly-sequenced genome. This protocol provides an extensive update to the previously published SCRMshaw protocol and aligns with the methods used in a recent annotation of over 30 insect regulatory genomes. It includes the most recent modifications to the SCRMshaw protocol and details an end-to-end pipeline that begins with a sequenced genome and ends with a fully-annotated regulatory genome. Relevant scripts are available via GitHub, and a living protocol that will be updated as necessary is linked to this article at protocols.io.

## Introduction

Although metazoan whole-genome sequences continue to accumulate at a rapid pace, comprehensive annotation of these genomes lags significantly behind. Particularly lacking is identification of non-coding regulatory sequences, which comprise a substantial fraction of the genome yet typically are completely lacking in available annotations. One reason why regulatory annotation so often lags behind genome sequencing is that historically, finding regulatory elements in the genome has been difficult even in well-studied model organisms because of their distant positions from target genes, the absence of a clear universal biochemical regulatory sequence marker, and the cell type specificity of regulatory element activity [1–4].

**Data Availability Statement:** All relevant data are within the manuscript and its Supporting Information files.

**Funding:** Funding for this work was provided by USDA grant 2019-67013-29354 to M.S.H. The

funders had no role in study design, data collection and analysis, decision to publish, or preparation of the manuscript.

**Competing interests:** The authors have declared that no competing interests exist.

Although both empirical and computational methods for regulatory element discovery have been developed (for review see [5–8]), a variety of limitations prevent them from being easily adopted for efficient annotation of newly-sequenced genomes. Empirical approaches are costly; can be difficult to validate depending on the availability of biological resources such as cell lines, antibodies, and tissue samples, or the existence of relevant technologies, such as transgenesis; and carry apparent false-positive and false-negative rates that can be surprisingly high (false-positive rates range as high as 40% for some ChIP-based methods [9–11] and from 10–20% for some ATAC-seq studies [12, 13]). Because regulatory elements may be functional only in certain cell types or under specific conditions, assays need to be applied to multiple tissues over many developmental stages and/or under varying environmental conditions in order to achieve comprehensive annotation. Although computational methods in theory avoid these issues, many approaches still rely on experimental data either for training or as input, which often negates their advantages. For non-model organisms, where only limited functional genomic data tend to be available, regulatory annotation is therefore often highly challenging.

We previously developed SCRMshaw, a supervised cis-regulatory module machine-learning method that shows strong performance in predicting transcriptional regulatory sequences ("enhancers") [14–19]. When trained on known enhancers from the fruit fly *Drosophila melanogaster*, SCRMshaw successfully predicts regulatory sequences across a wide range of insect species [17, 19–21]. Importantly, SCRMshaw requires as input only a sequenced genome and a basic gene annotation, making it highly suitable for producing preliminary regulatory annotations rapidly following initial publication of new genome assemblies. While benchmarking enhancer discovery methods is complicated due to the fact that there are no established true positive/true negative data sets against which to compare competing approaches [14], success rates from SCRMshaw appear to be on a par with or better than those from other rigorously-evaluated methods. Importantly, while we expect that other sequence-based enhancer discovery methods will, similar to SCRMshaw, be capable of cross-species discovery, only SCRMshaw so far has a track record of success in the cross-species setting.

We present here a detailed updated SCRMshaw protocol that replaces the earlier protocol published in 2019 [16] and aligns with the procedures described in our recent publication presenting the regulatory annotation of over 30 insect genomes [19]. The current protocol incorporates changes to the original SCRMshaw protocol as described by Asma and Halfon [14] and Asma et al. [19]. It includes additional steps developed by Asma et al. [19] to implement an end-to-end pipeline that begins with a sequenced genome and ends with a fully-annotated regulatory genome formatted for easy comparison to other SCRMshaw-annotated genomes.

The SCRMshaw pipeline consists of four basic parts (Fig 1). A pre-processing step checks the input files for proper formatting and removes minor unmapped/unannotated scaffolds from the genome. The SCRMshaw algorithm itself, which consists of three underlying statistical methods, is then run to scan the genome for high-scoring sequences. This step is dependent on training data, which can be downloaded from our GitHub site. Custom training sets can also be developed by the user; for instructions on custom training set generation, we refer the reader to Kazemian and Halfon [16]. The basic protocol described here is for "SCRMshaw_HD" [14], which requires use of a high-performance computing cluster and a minimum of 25 available nodes. However, SCRMshaw can also be run in its original lighter-weight form on a desktop computer by following the relevant protocol notes and/or referring to [16]. The third step in the SCRMshaw pipeline is post-processing, which determines the final set of enhancer predictions based on the SCRMshaw scores. Details on this step can be found in [19] and are diagrammed in Fig 2. The final step in the SCRMshaw pipeline, which is optional, maps putative enhancer target genes to their *Drosophila* orthologs. We find that this

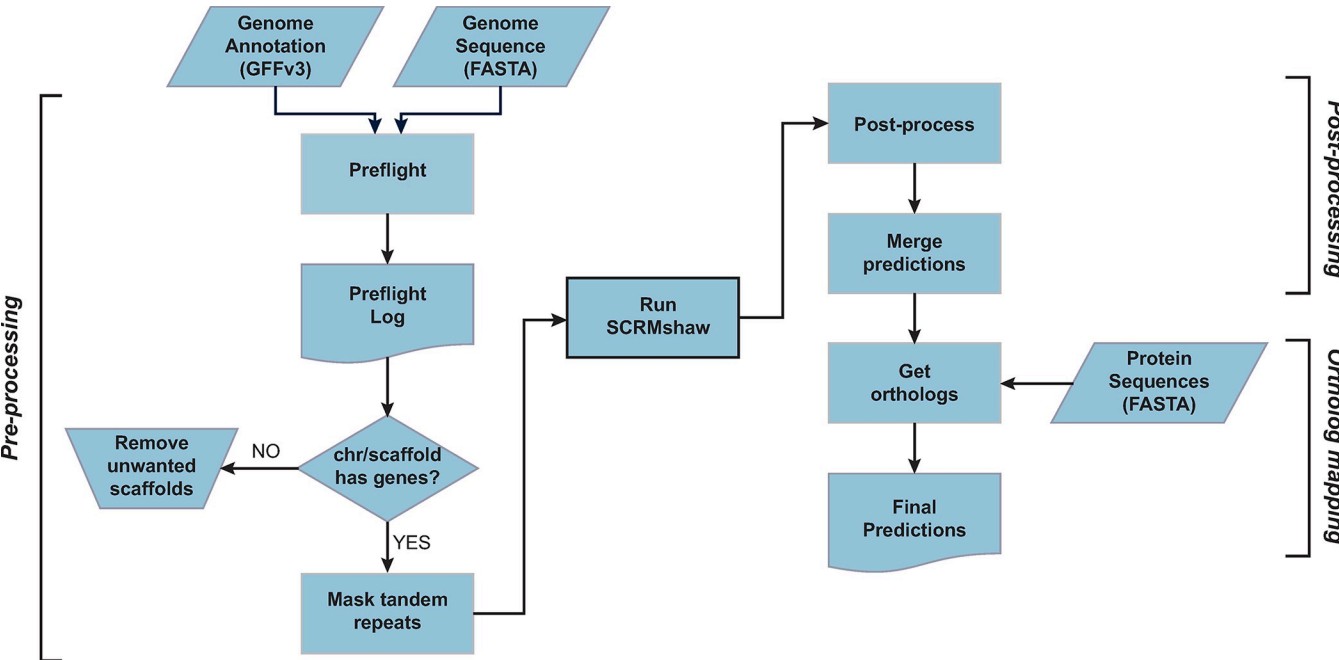

**Fig 1. The SCRMshaw pipeline.** The left side shows pre-processing steps, the right side, post-processing. Input to SCRMshaw consists of the genome sequence and gene annotation. A protein sequence annotation is supplied later for the ortholog mapping step. Adapted from [19].

step is helpful in that it provides recognizable gene names to what are otherwise frequently arbitrary designations, and enables comparison of locus-by-locus results between SCRMshaw outputs from multiple species. Note that target gene assignments are based solely on nearest-gene relationships, which is a simple but often non-accurate method of determining enhancer targets. Users may want to incorporate additional information into their analysis if accurate enhancer-target relationships are a primary goal of their study.

## Materials and methods

The protocol described in this peer-reviewed article is published on *protocols.io* (doi:10.17504/protocols.io.e6nvw1129lmk/v2) and is included for printing as S1 File with this article.

## Expected results

The SCRMshaw pipeline described in the accompanying protocol (S1 File) was used most recently to produce the 33 genome annotations reported by Asma et al. [19]. The complete set of scripts and SCRMshaw software from that study is available for download at our GitHub site (https://gitub.com/HalfonLab), and the annotation data can be searched at REDfly (http://redfly.ccr.buffalo.edu) or downloaded from Dryad (https://doi.org/10.5061/dryad.3j9kd51t0).

The SCRMshaw_HD pipeline can analyze an average-sized insect genome using several training sets in a matter of hours; we are able to run all but the largest or most fragmented genomes with our full default set of 48 training sets in under 72 hours. The bulk of the computational time is spent in the SCRMshaw step. To decrease run times, chromosomes and/or training sets can be split out and run as separate instances on additional sets of 25 nodes, if available, as a simple parallelization strategy. Storage space increases with genome size, mostly due to the larger number of kmers that must be stored, and can grow to several TB with larger genomes. However, the majority of this space can be released upon completion of the pipeline

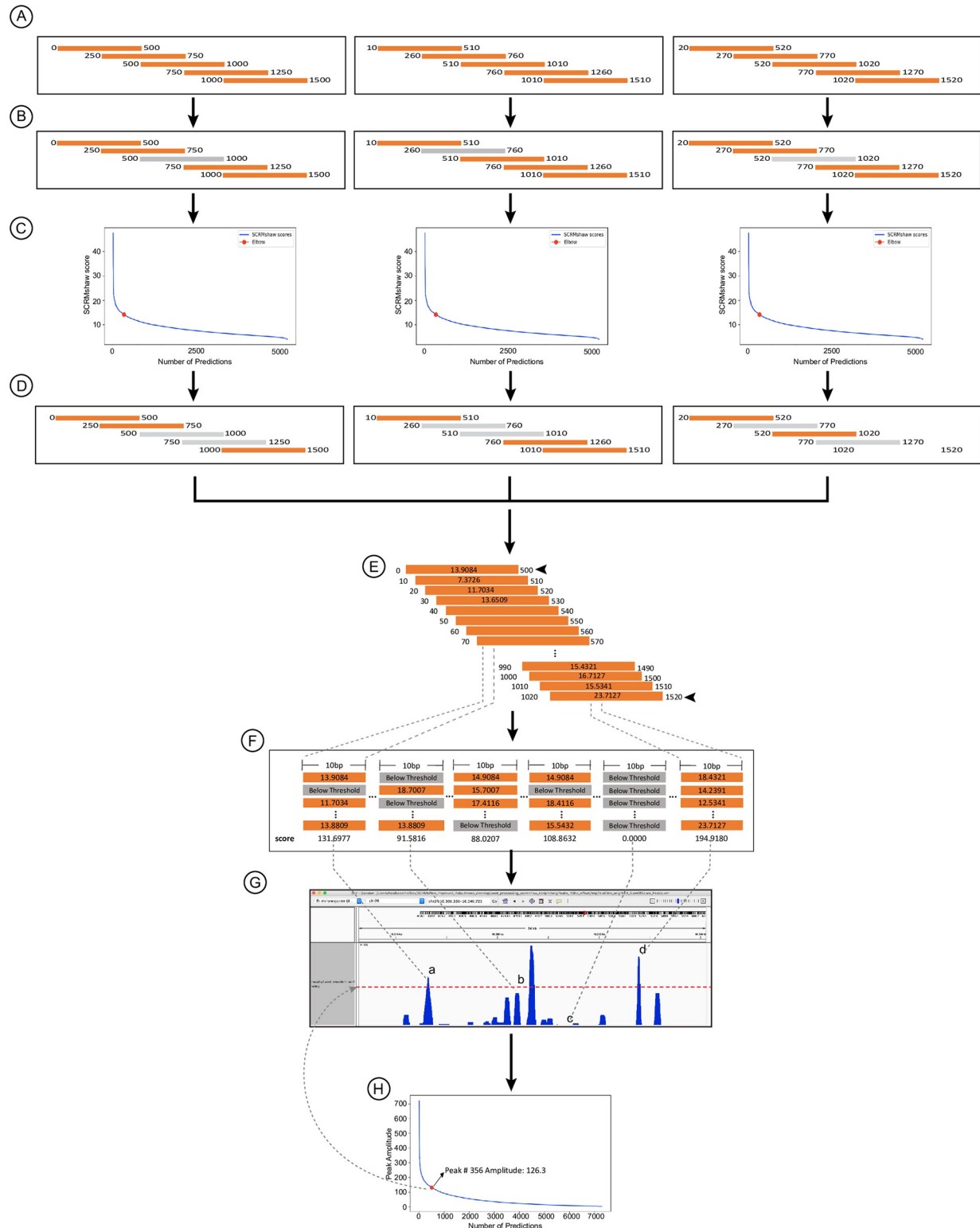

**Fig 2. Post-processing.** Post-processing is described in [14], with modifications in [19]. (A) Scores from each of the 25 individual SCRMshaw instances are assessed and (B) any 500 bp window whose score is below the value of the 5000[th] ranked score is eliminated by having its score reset to zero (B). (C) The "elbow" point of the SCRMshaw score curve of the 5000 top scores from each instance is then determined, and (D) any scores below the elbow point are reset to zero. (E, F) After these two rounds of score evaluation, windows are grouped together and (G) subjected to peak calling on 10 bp intervals. (H) Final predictions are chosen as peaks with an amplitude above the elbow point of the amplitude curve, represented by a red dot. Adapted from [19].

by deleting intermediate and temporary files, using "cleanup" scripts we have made available. If sufficient temporary storage space is not available, it is advisable to run the original light-weight SCRMshaw [16] rather than SCRMshaw_HD, which will keep storage requirements below 100GB for most genomes.

Each step of the pipeline produces output. An example of the output from the *preflight* step is provided in S2 File. *Preflight* validates the format of the input files and produces a comprehensive log file that highlights any issues along with basic information such as the number of chromosomes/scaffolds and their sizes, data types present in the annotation (e.g., 'gene', 'exon', 'ncRNA', etc.), and average intergenic distances. *Preflight* also identifies minor scaffolds that are not annotated as containing genes, which can then be discarded.

Output from SCRMshaw itself is described in detail in [16]. When using the preferred *SCRMshaw_HD* process, this is mainly intermediate output that is used for subsequent steps but not directly evaluated.

*Post-processing* generates two principal files. *scrmshawOutput_offset_0to240.bed* contains the top 5000 raw results from each of the multiple individual SCRMshaw instances run by *SCRMshaw_HD*. The post-processing script then uses these raw results to make final enhancer predictions, which it outputs in the *peaks_AllSets.bed* file. This file can be used as SCRMshaw output for downstream analysis and differs from the final output described below only in that putative target genes are not mapped to their *Drosophila* orthologs. If that final step is not conducted, it is recommended to sort and merge the results using BEDTools [22] before considering the analysis completed, as there may be overlap in enhancer predictions from different training sets or statistical models.

Final results are obtained following the ortholog mapping procedure and merging of overlapping results. An example of final output is provided in S3 File. The final output is in the form of an 18-column tab-delimited file organized as follows:

1. Chromosome

2. Start coordinate

3. End coordinate

4. Peak amplitude

5. SCRMshaw score

6. Flanking gene

7. *D. melanogaster* ortholog of flanking gene

8. Distance of hit from flanking gene (basepairs)

9. Location of hit relative to flanking gene

10. Local rank

11. Next closest flanking gene

12. *D. melanogaster* ortholog of next flanking gene

13. Distance of hit from flanking gene (basepairs)

14. Location of hit relative to flanking gene

15. Local rank

16. Training set

17.  Method (hexmcd, imm, pac)

18.  Rank

If the orthologous gene is not known, it is listed as "No_OrthoPara." Where predictions are merged, multiple results may be provided in each column, depending on the results of the merge (e.g., for method, "imm, hexmcd"). Peak amplitude, score, and rank will contain the best value from among the merged predictions. "Local rank" is described in [17], although its utility as a metric when using the SCRMshaw_HD post-processing procedure has not been determined.

SCRMshaw's performance relies heavily on training set quality. Based on various measures over many individual studies [inter alia 14, 15, 17–20, 23], we estimate that true-positive rates for enhancer prediction range from 50–85%, with most training sets reaching or exceeding 70%. We plan to continue to make additional and improved training data available via our GitHub training set repository (https://github.com/HalfonLab/dmel_training_sets). The protocol linked to this article (S1 File) exists as a living protocol at *protocols.io* (http://dx.doi.org/10.17504/protocols.io.e6nvw1129lmk/v2), and we will continue to update it as improvements to the SCRMshaw pipeline are developed. Aspects of SCRMshaw under continued development include investigating optimal repeat masking strategies for genomes with different types and degrees of repeats, and how best to combine and weight the scores from the three individual SCRMshaw scoring methods of IMM, HexMCD, and PAC-rc. Users should bear in mind that SCRMshaw predictions are, ultimately, predictions, and appropriate validation experiments should be undertaken for any sequences of particular interest.

## Supporting information

**S1 File. Step-by-step protocol, also available on protocols.io.**
(PDF)

**S2 File. Example of the output from the "preflight" step run on the *Apis mellifera* (honeybee) genome.**
(PDF)

**S3 File. Example final output for SCRMshaw_HD run on the *Apis mellifera* genome using the "mapping2.wing" training set.**
(BED)

## Acknowledgments

We thank members of the Halfon lab for helpful comments on the protocol and the manuscript.

## Author Contributions

**Conceptualization:** Hasiba Asma, Marc S. Halfon.

**Funding acquisition:** Marc S. Halfon.

**Investigation:** Luna Liu.

**Methodology:** Hasiba Asma, Marc S. Halfon.

**Software:** Hasiba Asma, Luna Liu, Marc S. Halfon.

**Supervision:** Marc S. Halfon.

**Writing – original draft:** Marc S. Halfon.

**Writing – review & editing:** Hasiba Asma, Luna Liu, Marc S. Halfon.

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
