## [Decision Letter · Decision Letter 0]

17 Sep 2024

PONE-D-24-23787SCRMshaw: Supervised cis-regulatory module prediction for insect genomesPLOS ONE

Dear Dr. Halfon,

Thank you for submitting your manuscript to PLOS ONE. After careful consideration, we feel that it has merit but does not fully meet PLOS ONE’s publication criteria as it currently stands. Therefore, we invite you to submit a revised version of the manuscript that addresses the points raised during the review process.

Add a little more into the opening paragraph, about the problem of predicting regulatory elements. And specifically, which other tools attempt the same as SCRMshaw? Mention a few benefits/drawbacks of SCRMshaw vs these tools, and cite a/some benchmarking papers on that.

Minor point

Line 108

Is this the best heading for this section? “Expected Results”

We look forward to receiving your revised manuscript.

Kind regards,

Arnar Palsson, Ph.D.

Academic Editor

PLOS ONE

Journal Requirements:

1. When submitting your revision, we need you to address these additional requirements. Please ensure that your manuscript meets PLOS ONE's style requirements, including those for file naming. The PLOS ONE style templates can be found at https://journals.plos.org/plosone/s/file?id=wjVg/PLOSOne_formatting_sample_main_body.pdf and https://journals.plos.org/plosone/s/file?id=ba62/PLOSOne_formatting_sample_title_authors_affiliations.pdf 2. We note that the grant information you provided in the ‘Funding Information’ and ‘Financial Disclosure’ sections do not match.  When you resubmit, please ensure that you provide the correct grant numbers for the awards you received for your study in the ‘Funding Information’ section. 3. Thank you for stating the following financial disclosure: "Funding for this work was provided by USDA grant 2019-67013-29354 to M.S.H." Please state what role the funders took in the study.  If the funders had no role, please state: "The funders had no role in study design, data collection and analysis, decision to publish, or preparation of the manuscript." If this statement is not correct you must amend it as needed. Please include this amended Role of Funder statement in your cover letter; we will change the online submission form on your behalf. 4. We note you have not yet provided a protocols.io PDF version of your protocol and/or a protocols.io DOI. When you submit your revision, please provide a PDF version of your protocol as generated by protocols.io (the file will have the protocols.io logo in the upper right corner of the first page) as a Supporting Information file. The filename should be S1_file.pdf, and you should enter “S1 File” into the Description field. Any additional protocols should be numbered S2, S3, and so on. Please also follow the instructions for Supporting Information captions [https://journals.plos.org/plosone/s/supporting-information#loc-captions]. The title in the caption should read: “Step-by-step protocol, also available on protocols.io.” Please assign your protocol a protocols.io DOI, if you have not already done so, and include the following line in the Materials and Methods section of your manuscript: “The protocol described in this peer-reviewed article is published on protocols.io (https://dx.doi.org/10.17504/protocols.io.[...]) and is included for printing purposes as S1 File.” You should also supply the DOI in the Protocols.io DOI field of the submission form when you submit your revision. If you have not yet uploaded your protocol to protocols.io, you are invited to use the platform’s protocol entry service [https://www.protocols.io/we-enter-protocols] for doing so, at no charge. Through this service, the team at protocols.io will enter your protocol for you and format it in a way that takes advantage of the platform’s features. When submitting your protocol to the protocol entry service please include the customer code PLOS2022 in the Note field and indicate that your protocol is associated with a PLOS ONE Lab Protocol Submission. You should also include the title and manuscript number of your PLOS ONE submission. 5. We note that your Data Availability Statement is currently as follows: "All relevant data are within the manuscript and its Supporting Information files." Please confirm at this time whether or not your submission contains all raw data required to replicate the results of your study. Authors must share the “minimal data set” for their submission. PLOS defines the minimal data set to consist of the data required to replicate all study findings reported in the article, as well as related metadata and methods (https://journals.plos.org/plosone/s/data-availability#loc-minimal-data-set-definition). For example, authors should submit the following data: - The values behind the means, standard deviations and other measures reported;- The values used to build graphs;- The points extracted from images for analysis. Authors do not need to submit their entire data set if only a portion of the data was used in the reported study. If your submission does not contain these data, please either upload them as Supporting Information files or deposit them to a stable, public repository and provide us with the relevant URLs, DOIs, or accession numbers. For a list of recommended repositories, please see https://journals.plos.org/plosone/s/recommended-repositories. If there are ethical or legal restrictions on sharing a de-identified data set, please explain them in detail (e.g., data contain potentially sensitive information, data are owned by a third-party organization, etc.) and who has imposed them (e.g., an ethics committee). Please also provide contact information for a data access committee, ethics committee, or other institutional body to which data requests may be sent. If data are owned by a third party, please indicate how others may request data access. 6. Please include captions for your Supporting Information files at the end of your manuscript, and update any in-text citations to match accordingly. Please see our Supporting Information guidelines for more information: http://journals.plos.org/plosone/s/supporting-information.

Additional Editor Comments:

This is a fine manuscript. Sorry for the delay, we were waiting on reviewer 2 that, eventually didnt come through.

Add a little more into the opening paragraph, about the problem of predicting regulatory elements. And specifically, which other tools attempt the same as SCRMshaw? Mention a few benefits/drawbacks of SCRMshaw vs these tools, and cite a/some benchmarking papers on that.

Minor point

Line 108

Is this the best heading for this section? “Expected Results”

Reviewers' comments:

Reviewer's Responses to Questions

**Comments to the Author**

1. Does the manuscript report a protocol which is of utility to the research community and adds value to the published literature?

Reviewer #1: Yes

2. Has the protocol been described in sufficient detail?

To answer this question, please click the link to protocols.io in the Materials and Methods section of the manuscript (if a link has been provided) or consult the step-by-step protocol in the Supporting Information files.

The step-by-step protocol should contain sufficient detail for another researcher to be able to reproduce all experiments and analyses.

Reviewer #1: Yes

3. Does the protocol describe a validated method?

Reviewer #1: Yes

4. If the manuscript contains new data, have the authors made this data fully available?

Reviewer #1: N/A

**5. Is the article presented in an intelligible fashion and written in standard English?**

Reviewer #1: Yes

6. Review Comments to the Author

Reviewer #1: 1. The manuscript describes an updated protocol of SCRMshaw, a computational tool designed to predict cis-regulatory modules in insect genomes. This updated version is of significant utility to the research community for several reasons. First, its minimal resource requirements make it accessible to a wide range of researchers, including those with limited computational skills and resources. Second, the flexibility of SCRMshaw to be tailored for specific needs adds considerable value to the tool. This update which includes a detailed pipeline, available via GitHub and the provision of a living protocol linked to the article further enhance its utility and relevance, ensuring that it can be easily adapted and updated by the research community.

2. The protocol provided in the Materials and Methods section of the manuscript and in protocols.io shows that the authors have meticulously detailed the steps involved in using SCRMshaw. The protocol covers all the necessary stages, from the initial preparation of the genome sequence to the final annotation of regulatory elements. Each step is accompanied by clear instructions and helpful screenshots, and the inclusion of scripts available via GitHub ensures that the users can easily implement the protocol. A great advantage of this detailed protocol, is the notes and recommendations for users with basic or optimal computational resources. Therefore, the protocol has been described in sufficient detail to allow their implementation by the research community.

3. The SCRMshaw protocol describes a validated method that has been tested in Halfon's lab and supported by scientific evidence developed since 2009. This foundation has triggered updated and effective versions of the protocol. The use of training data from well-characterized Drosophila regulatory sequences, and their application in the annotation of 33 insect regulatory genomes, demonstrate its robustness and reliability across multiples datasets and species. Notably, the genome annotation of the most relevant Orders in the Insecta class, such as Hemiptera, Hymenoptera, Coleoptera, Lepidoptera and Diptera, makes this tool a useful reference for searching cis-regulatory sequences in both model and non-model organisms currently studied by the research community. The detailed description of the end-to-end pipeline, from genome sequencing to regulatory annotation, further supports the validity of the method.

4. The manuscript is well-written in standard English. In general, the language used is precise and appropriate for a scientific audience, clearly describing the purpose and advantages of the SCRMshaw tool and presenting a logical and coherent flow of information.

Suggestions for improvement:

-The authors should include a brief discussion on the limitations of the SCRMshaw tool and how these might be addressed in future versions and updates. This would provide a balanced view and help researchers understand the tool's scope.

7. PLOS authors have the option to publish the peer review history of their article (what does this mean?). If published, this will include your full peer review and any attached files.

Reviewer #1: **Yes: **Keity J. Farfán-Pira

---

## [Author Response · Author response to Decision Letter 0]

20 Sep 2024

Response to Review:

We thank the Reviewer and the Editor for their comments on our manuscript. We have made the following changes in response:

1. We added significant new text in the Introduction discussing the general issues surrounding regulatory element discovery, and referenced several review articles that go into detail about various experimental and computational discovery methods. Given that this is a Protocols paper and that we and others have reviewed these methods previously, we did not think it desirable to provide an extensive treatment here. However, the text and cited articles will now allow the interested reader to easily discover more detailed information.

2. Benchmarking regulatory element discovery algorithms is tricky for a large number of reasons, including lack of any true “truth” data sets to benchmark against, and frequently different requirements for input and/or training for the leading approaches. We comment on some of these issues in our earlier paper Asma and Halfon 2019 (PMID: 30953451), which we now cite on line 95. Previous papers on SCRMshaw, which are cited here, do have some benchmarking to the extent possible. Importantly, no good comparisons exist for attempting cross-species discovery, i.e., training on one species and conducting predictions on another, which is the main focus and strength of SCRMshaw. We have added comments on this at lines 93-99 and point out that based on published results “success rates from SCRMshaw appear to be on a par with or better than those from other rigorously-evaluated methods.”

3. We added additional text about SCRMshaw’s limitations at lines 211-218. We also provide a clear reminder that “SCRMshaw predictions are, ultimately, predictions, and appropriate validation experiments should be undertaken for any sequences of particular interest.”

4. With respect to the Editor’s comment “Line 108 [now line 136]: Is this the best heading for this section? ‘Expected Results’”, we followed the template provided by PLoS One for “Protocols” papers, which specifies the section headings. We would be happy to substitute this for a different heading at the Editor’s suggestion.

---

## [Editor Report · Decision Letter 1]

25 Sep 2024

SCRMshaw: Supervised cis-regulatory module prediction for insect genomes

PONE-D-24-23787R1

Dear Dr. Halfon,

We’re pleased to inform you that your manuscript has been judged scientifically suitable for publication and will be formally accepted for publication once it meets all outstanding technical requirements.

Kind regards,

Arnar Palsson, Ph.D.

Academic Editor

PLOS ONE
---

## [Editor Report · Acceptance letter]

27 Sep 2024

PONE-D-24-23787R1 

PLOS ONE

Dear Dr. Halfon, 

I'm pleased to inform you that your manuscript has been deemed suitable for publication in PLOS ONE. Congratulations! Your manuscript is now being handed over to our production team.

Kind regards, 

on behalf of

Dr. Arnar Palsson 

Academic Editor

PLOS ONE